# Correlations between Molecular Alterations, Histopathological Characteristics, and Poor Prognosis in Esophageal Adenocarcinoma

**DOI:** 10.3390/cancers15051408

**Published:** 2023-02-23

**Authors:** Arianna Orsini, Luca Mastracci, Isotta Bozzarelli, Anna Ferrari, Federica Isidori, Roberto Fiocca, Marialuisa Lugaresi, Antonietta D’Errico, Deborah Malvi, Erica Cataldi-Stagetti, Paola Spaggiari, Anna Tomezzoli, Luca Albarello, Ari Ristimäki, Luca Bottiglieri, Kausilia K. Krishnadath, Riccardo Rosati, Uberto Fumagalli Romario, Giovanni De Manzoni, Jari Räsänen, Giovanni Martinelli, Sandro Mattioli, Elena Bonora

**Affiliations:** 1Department of Medical and Surgical Sciences (DIMEC), Alma Mater Studiorum, University of Bologna, 40126 Bologna, Italy; 2Unit of Anatomic Pathology, Ospedale Policlinico San Martino IRCCS, 16125 Genova, Italy; 3Department of Surgical Sciences and Integrated Diagnostics (DISC), University of Genova, 16125 Genova, Italy; 4IRCCS Istituto Romagnolo per lo Studio dei Tumori (IRST) “Dino Amadori”, 47014 Meldola, Italy; 5Division of Thoracic Surgery, Maria Cecilia Hospital, GVM Care & Research Group, Cotignola, 48022 Ravenna, Italy; 6Pathology Unit, IRCCS Azienda Ospedaliero-Universitaria di Bologna, 40126 Bologna, Italy; 7Department of Experimental, Diagnostic and Specialty Medicine (DIMES), Alma Mater Studiorum, University of Bologna, 40126 Bologna, Italy; 8Unit of Anatomic Pathology, Humanitas University, 20089 Milan, Italy; 9Unit of Anatomic Pathology, Azienda Ospedaliera di Verona, 37122 Verona, Italy; 10Pathology Unit, San Raffaele Scientific Institute, 20135 Milan, Italy; 11Department of Pathology, HUSLAB and HUS Diagnostic Center, University of Helsinki, 00170 Helsinki, Finland; 12Helsinki University Hospital, 00170 Helsinki, Finland; 13Unit of Anatomic Pathology, Istituto Europeo di Oncologia, 20122 Milan, Italy; 14Laboratory of Experimental Medicine and Pediatrics (LEMP), Department of Gastroenterology and Hepatology, University Hospital Antwerp, 2650 Antwerp, Belgium; 15Department of Gastrointestinal Surgery, San Raffaele Hospital, Vita-Salute San Raffaele University, 20135 Milan, Italy; 16Digestive Surgery, European Institute of Oncology, IRCCS, 20122 Milan, Italy; 17Department of Surgery, General and Upper G.I. Surgery Division, University of Verona, 37126 Verona, Italy; 18Department of General Thoracic and Esophageal Surgery, Helsinki University Hospital, 00170 Helsinki, Finland

**Keywords:** esophageal adenocarcinoma, *TP53*, *HNF1alpha*, *SMAD4*

## Abstract

**Simple Summary:**

The molecular heterogeneity of esophageal adenocarcinoma (EAC), a severe malignancy with increasing incidence and low survival rates, misperceives the underlying biology of tumor onset and development. However, advances in high-throughput next-generation sequencing (NGS) technologies have highlighted the potential role of somatic DNA sequence markers for new diagnostic techniques or constitute novel therapeutic targets. Thus, in order to identify a molecular and prognostic signature in EAC patients, we decided to integrate the sequencing of specimens from naïve patients (not treated with chemo-radiotherapy) with histological classification, with the aim of identification of potential biomarkers, and patient stratification. Combining different approaches paves the way for early identification and the selection of better therapy.

**Abstract:**

Esophageal adenocarcinoma (EAC) is a severe malignancy with increasing incidence, poorly understood pathogenesis, and low survival rates. We sequenced 164 EAC samples of naïve patients (without chemo-radiotherapy) with high coverage using next-generation sequencing technologies. A total of 337 variants were identified across the whole cohort, with TP53 as the most frequently altered gene (67.27%). Missense mutations in TP53 correlated with worse cancer-specific survival (log-rank p = 0.001). In seven cases, we found disruptive mutations in HNF1alpha associated with other gene alterations. Moreover, we detected gene fusions through massive parallel sequencing of RNA, indicating that it is not a rare event in EAC. In conclusion, we report that a specific type of TP53 mutation (missense changes) negatively affected cancer-specific survival in EAC. HNF1alpha was identified as a new EAC-mutated gene.

## 1. Introduction

Esophageal adenocarcinoma (EAC) is a severe malignancy with increasing incidence in Western countries over the past few decades and a relatively high mortality since overall prognosis remains bleak and the 5-year survival rate is just 35–45% [1].

EAC develops from the cells that release mucus and other fluids and may arise according to the widely accepted sequence gastroesophageal reflux disease (GERD) /intestinal metaplasia/dysplasia/ adenocarcinoma [2].

In clinical practice, complete endoscopic evaluation of GERD symptoms includes evaluation of erosive esophagitis and its complication and inspection for BE with multiple biopsies when present [3]. BE is relatively common in the general population, with a 1–2% prevalence (up to 10% in those with reflux symptoms) [4], but only about 1% of patients progress to cancer each year [5]. The ability to treat pre-invasive, dysplastic lesions with endoscopic resection and/or ablation instead of more extensive and invasive procedures such as chemotherapy with or without radiotherapy and esophagectomy, which have associated co-morbidities and poor 5-year survival rates, makes early diagnosis highly clinically relevant [6].

In a recent study on a large series of EAC cases submitted to surgery (without neoadjuvant treatment), a diagnostic algorithm which separated adenocarcinomas with glandular architecture from other rare histotypes, and further graded the former and subtyped the latter, was adopted [7]. This morphologic distinction has proven to have a significant prognostic impact on its own or dichotomized into lower and higher risk carcinomas, especially when coupled with stage. Indeed, the stage plus histotype combination showed a high discriminating power for 5-year cancer-specific survival, ranging from 87.6% in the stage II lower risk group to 14% in the stage IVA higher risk group [7]. Given the histological differences observed in EAC, it is of great interest to investigate the underlying biology of the tumor and to understand the molecular alterations correlated with those distinctive patterns which provide strong prognostic factors. Indeed, several genomic studies have included EAC in a group of tumors with one of the most frequent rates of copy number alterations (CNAs), somatic structural rearrangements, and elevated mutation frequency, with different mutational signatures and epigenetic mechanisms giving rise to significant inter and intra-tumor heterogeneity [8]. Large-scale sequencing studies have revealed distinct mutational signatures in EAC, and the presence of multiple CNAs was possibly correlated to a worse outcome; however, an exhaustive correlation with clinical outcomes and specific histotypes has not yet been provided. In a large genomic study, Secrier et al. identified three distinct molecular subtypes of molecular signatures: (i) an enrichment for BRCA signature with prevalent defects in the homologous recombination pathway; (ii) a dominant T > G mutational pattern associated with a high mutational load and neoantigen burden; and (iii) a C > A/T mutational pattern with evidence of an aging imprint. However, the clinical characteristics of the three subgroups did not differ significantly [9].

In EAC, a number of potential driver mutations have been described, with many of the mutations occurring in tumor suppressor genes (e.g., *TP53*, *SMAD4,* and *ARID1A*). The overall picture is of genomic instability and significant heterogeneity between patients. The accumulation of structural variants appears to be a gradual process throughout the disease’s natural history [6,10]. According to the Cancer Genome Atlas [8], EACs contain molecular changes similar to the chromosomal instability (CIN) subtype of stomach cancer. This suggests that EAC treatment may be improved by grouping them with CIN stomach cancers. Genomic alterations may also represent effective targets for therapy, i.e., frequent alterations of genes that regulate the cell cycle may be treated with existing drugs. Moreover, one-third of the EACs studied harbor an alteration to the *ERBB2* gene, which encodes the HER2 protein and may be targeted with HER2 inhibitors. However, the overall genomic heterogeneity and the rearrangements occurring during tumor progression make it difficult to define valuable prognostic molecular signatures underlying biology that drive tumor onset and development [11].

We coupled a deep sequencing analysis of the most commonly mutated genes in EAC with the recent novel classification based on histological subtypes [7] in order to stratify patients based on clinical and molecular features for better personalized care. We evaluated the genetic alterations of cancer-related genes that we found recurrently mutated in a previous study on a small group of EAC cases [12]. Herein we analyzed specimens of 164 naïve patients (without chemo-radiotherapy and not subjected to neoadjuvant treatments) using next-generation sequencing approaches. Moreover, we characterized the presence of novel gene fusion transcripts, as markers of genomic alterations, in a number of these cases for which we had RNA available from tumor specimens. Molecular data were correlated with the histological subtypes and the clinical outcomes.

## 2. Materials and Methods

### 2.1. Sample Recruitment

The DNA samples were extracted from the surgical samples obtained at the following centers: Istituto Europeo di Oncologia (IEO), Milano, and IRCCS Ospedale San Raffaele, Milano, Ospedale di Verona, Verona, for a total of 164 samples. The inclusion criteria consisted of the presence of adenocarcinoma of the esophagogastric junction; no neoadjuvant treatment (chemo-radiotherapy-naïve EACs); and full clinical medical history and follow-up up to 60 months after surgery. All of the surgical resections were formalin-fixed paraffin-embedded (FFPE), re-evaluated by gastrointestinal pathologists, and classified according to the EACSGE histological classification [7]. The cases analyzed for mutation analysis of *TP53* only have been previously described [12] and they were re-classified according to the EACSGE classification.

### 2.2. Custom EAC Panel: Library Preparation, Hybridization, Sequencing, and Bioinformatic Analysis

DNA was extracted from two 40 μm-thick, FFPE sections using a QIAMP DNA FFPE Tissue Kit (Cat. 56404; Qiagen, Hilden, Germany) according to the manufacturer’s protocol. Dual-index paired-end libraries were prepared using a Lotus DNA library prep kit (Cat. 10001074; Integrated DNA Technologies IDT Inc., Coralville, IA, USA) according to the manufacturer’s instructions. The protocol followed three major steps: an enzymatic preparation (with fragmentation to obtain 300–350 bp DNA fragments) where end-repair and dA-tailing were performed, the ligation of stubby adapters was performed, and PCR amplification for 11 cycles with indexing primers was performed (to incorporate sample-unique indexing sequences and P5 and P7 sequences to attach to the flow-cell). After purification, the single DNA libraries were run on 3% agarose gel to confirm the appropriate size and quantified using a Qubit dsDNA BR Assay Kit (Cat. Q33265; ThermoFisher Scientific, Vilnus, Lituania). Five hundred ng of each library preparation was pooled into groups of 16 samples to perform hybridization and enrichment for selected gene regions. This step was performed using an xGen Lockdown probe pool and an xGen hybridization capture of DNA libraries kit (IDT), according to the protocols. Each pool of 16 samples was hybridized to the capture probes for 16 h at 65 °C. xGen Lockdown Probes were individually synthesized, including 5′ biotinylated oligos, and were assembled in a custom panel of 26 genes for target capture. The genes selected for this study are listed in Figure 1. The hybridized regions were then captured with streptavidin magnetic beads and, after non-bound products’ removal, a post-capture PCR of 11 cycles was performed. The enriched library pools were checked for quantity and size with a Qubit dsDNA HS Assay kit (Cat. Q33230; Thermo Fisher Scientific, Waltham, MA, USA) and 2100 Bioanalyzer High Sensitivity DNA (Agilent Technologies, Santa Clara, CA, USA), respectively. Each pool was normalized to 1.3 pM and then sequenced on an Illumina NextSeq 500 platform (Illumina San Diego, CA, USA) at 150 bp paired ends. Data analysis was performed with an in-house pipeline [12]. In particular, Fastq files containing raw reads were checked using FastQC (https://www.bioinformatics.babraham.ac.uk/projects/fastqc/ v.0.11.8) and aligned using BWA (bio-bwa.sourceforge.net v.0.7.17-r1188) to the human reference (hg19). PCR-duplicated reads were marked and removed using Picard. Putative somatic variants, including SNPs and small insertions/deletions (indels), were identified using GATK software (software.broadinstitute.org/gatk/ v.4.0.10). The raw mutation calls were filtered to exclude false calls based on base quality, allele frequency of mismatched bases, and possible occurrences of strand bias. The identified mutations were further annotated and prioritized with Ensembl VEP (www.ensembl.org/Tools/VEP v.94).

### 2.3. RNA Analysis

RNA was extracted from the FFPE samples (five for each EACSGE subgroup). The samples were then selected among the ones with good quality RIN scores and DV200 > 40%. Starting from 100 ng of 27 RNA, libraries were prepared using a TruSight RNA Pan-Cancer Panel Kit (Illumina, San Diego, CA, USA; 1385 cancer-associated genes), following the manufacturer’s protocol. On twenty-two of the libraries that passed the protocol quality checks, paired-end RNA sequencing was performed (Reagent Kit v3-150 cycles, MiSeq, Illumina, San Diego, CA, USA) and raw sequencing data were converted to FASTQ file format and analyzed by combining FusionCatcher (FC(1)), STAR-Fusion (SF), and two Basespace applications [RNA-Seq Alignment v.1.1.0 (RSA) and TopHat Alignment v.1.0.0 (THA); Illumina, San Diego, CA, USA]. The reference Homo sapiens UCSC hg19 (RefSeq and Gencode gene annotations) was used for all the aligners.

We retained the fusions detected by at least three tools and we introduced further criteria to retain or reject fusions detected by two tools or one tool [see PCT application No. PCT/EP2021/065692 (10 June 2021): Method to identify linked genetic fusions]. The gene fusions were confirmed with Sanger sequencing, as previously described [12].

### 2.4. Immunohistochemistry Analysis

Immunohistochemistry (IHC) analysis was performed automatically with a Benchmark XT^®^ immunostainer (Ventana Medical Systems) HNF1alpha antigen. The immunohistochemical analysis was validated through positive controls (as an external positive control put on the slide according to Bragoni et al. [13]) and negative controls (by omitting the primary antibody). Cases carrying predicted damaging variants in HNF1alpha and cases with no variants in the gene were evaluated by IHC from FFPE surgical specimens. IHC was performed for HNF1alpha and scoring was carried out by two independent expert pathologists, blindly with respect to the mutation status.

IHC analysis for SMAD4 and p53 was carried out as described previously [12]. Evaluation of SMAD4 immunostains was performed by two expert pathologists. For each case, the percentage of neoplastic cells with SMAD4 preserved or lost immunosignal was collected.

### 2.5. Statistical Analysis

The χ^2^ test or Fisher’s test (an expected number less than five) and the Mann–Whitney test were used to analyze categorical and continuous variables, respectively. The correlations were analyzed with Spearman’s rho coefficient. Survival analysis was performed using the Kaplan–Meier method and the log-rank test. *p*-values < 0.05 were considered significant. Data were analyzed using SPSS (version 15.0) (SPSS Inc., Chicago, IL, USA) and Prism (GraphPad Software Inc., San Diego, CA, USA). Univariate and multivariate (forward stepwise conditional method) Cox regression analyses were performed to estimate the effects of clinical, genetic, and pathological parameters on CSS. In the stepwise procedure, significance levels of 0.05 for entering and 0.10 for removing the respective explanatory variables were used to determine the independent risk factors. For the power calculations, we used G*Power version 3.1.9.6 [14].

## 3. Results

### 3.1. Genetic Alterations Identified in the EAC Samples

A total of 337 variants were identified across the whole cohort of 164 EAC cases (Figure 1A). All of the FFPE samples achieved good sequence representation with average coverage among samples of 700×. Examples of identified mutations are reported in Figure 1B.

Point mutations were the most frequent variations (82.21%), followed by insertions and deletions (17.78%). *TP53* was the most frequently altered gene, with 110/164 cases carrying at least one mutation in this gene (67.1%). Out of the 110 variants in the *TP53* gene, there was a prevalence in missense (77) vs. loss of function (LOF, including premature stop codons, splice site alterations, and frameshift variants) (33) variants. Five cases carried two variants in *TP53*. Most of the mutations have already been reported in the “TP53 database” (https://tp53.isb-cgc.org/ accessed on 10 October 2022) and were functionally analyzed: 76 out of 77 missense changes (98.7%) in *TP53* found in our EAC cohort were functionally damaging (Appendix A).

Alterations in other genes occurred at a lower frequency, with *ATM* (18%), *MSH6* (11%), *PI3KCA* (9%), *APC* and *SMAD4* (8%), and *CDKN2A* and *SMARCA4* (7%) being the most frequently hit genes. The majority of the samples carried concurrent variants in different genes (Figure 1A).

### 3.2. HNF1alpha Mutations in EAC

We found seven variants in the *HNF1alpha* gene, encoding for a tumor suppressor protein; in seven cases (Figure 1C), the gene that we previously found mutated in a small number of EAC cases [12]. The mutations mapped to the DNA binding domain (three single nucleotide substitutions and one frameshift indel variant) and to the transactivation domain (two single nucleotide substitutions and one frameshift indel variant) of *HNF1alpha*. The missense variants were predicted to be damaging according to the prediction program MCAP (MCAP score > 0.7; http://bejerano.stanford.edu/mcap/ accessed on 15 November 2022) [15]. In six out of sevem cases, *HNF1alpha* variants occurred in association with other gene alterations (Figure 1A).

We evaluated via IHC analysis the protein expression profiles of the available samples carrying the different *HNF1alpha* variants vs. a case without mutations (the control). Compared to the control sample, decreased staining was observed in the patients with *HNF1alpha* damaging variants. The decrease in staining correlated to an increased frequency of the variant alleles in the tumor, as detected by NGS data analysis (Figure 1D and Appendix A).

### 3.3. Correlation of Variants in Different Genes

As already reported in many studies, we found a variety of genetic alterations in the 164 samples analyzed via NGS, with *TP53* being the most mutated gene. To evaluate whether mutations co-occurred significantly in specific genes, we performed a correlation analysis between the damaging variants identified in the different oncology-related genes, using Spearman’s rho coefficient. We identified several statistically significant positive correlations between the presence of mutations in specific genes, as well as several negative correlations, as reported in Appendix A. *TP53* mutations, as an example, correlated positively with *CDKN2A* (0.162, *p* = 0.035) but correlated negatively with *ATM* (−0.147, *p* = 0.047), *HNF1alpha* (−0.166, *p* = 0.031), and *MET* (−0.195, *p* = 0.011). *HNF1alpha* on the other hand, correlated positively with *PIK3CA* (0.248, *p* < 0.001), *CTNNB1* (0.281, *p* < 0.001), and *RET* (0.161, *p* = 0.034), suggesting that specific genes were concurrently mutated in these tumors. We also assessed in the COSMIC project COSU535, containing data on 409 EAC cases, and the presence of co-occurring variants between genes and found mutations of *TP53* and *APC* (21/409, 5%), *TP53* and *CDKN2A* in 25/409 (6%), and *TP53* and *SMAD4* in 30/409 (7%) in the same samples, reinforcing the concept of the genetic heterogeneity of EAC.

### 3.4. Evaluating Associations between Genetic Variants and Histopathological and Clinical Phenotypes

In order to connect the presence of genetic variants with clinical and/or morpho-functional characteristics, we performed further analyses taking into account cancer-specific survival (CSS), recurrence, and the EACSGE classification, introduced by Fiocca et al. [7]. This classification was based on morphological features of esophageal/esophagogastric junction adenocarcinoma, which divided the cases into two main categories with a different prognose: lower risk, including glandular well differentiated (GL WD), mucinous muconodular carcinoma (MMC), and diffuse desmoplastic (DDC) subgroups; and higher risk, including glandular poorly differentiated (GL PD), diffuse anaplastic (DAC), invasive mucinous carcinomas (IMC), and mixed (MIX) subgroups.

This analysis provided significant data only for *TP53*, since the number of mutations in this gene was high enough to allow a statistical association to different parameters. 

We estimated the clinical outcomes in relation to the presence of *TP53* variants and to the specific types of *TP53* variants, i.e., missense variants or LOF variants. Poor survival and recurrence were significantly associated with *TP53* mutations (*p* = 0.039; Appendix A; *p* = 0.031, Appendix A, respectively). Considering the EACSGE classification in the lower risk and higher risk groups, the presence of *TP53* mutations and the higher risk group were significantly associated (*p* = 0.022; Appendix A).

These data prompted us to extend the analysis to include additional EAC cases for which we had genetic material and the *TP53* mutation status, the clinical parameters, and the morphological classification according to EACSGE for a total of 202 individuals. The overall results are presented in Figure 2A,B and Appendix A. We could show that cancer-specific survival was negatively affected by the presence of missense mutations in the higher risk cases (*p* = 0.001, Appendix A), as also shown by Kaplan–Meier curve analysis (Figure 2A). Considering the different subclasses of the higher and lower risk groups, we could observe how the statistical association was mainly driven by the GD-PD classes in the presence of missense variants in the *TP53* gene (*p* = 0.001; Appendix A and Figure 2B).

*TP53* mutations and age also showed a significant association, as reported in Appendix A (*p* = 0.029, Kruskal–Wallis test).

We investigated whether the presence of the different types of *TP53* variants (missense and LOF) could be detected by immunohistochemical analysis of the tumor specimens. We evaluated the presence of different types of variants and the staining pattern observed for p53 in terms of overexpression or loss of staining (Appendix A), according to our previously reported methods [12], in which missense variants were associated with p53 staining. We found a significant association between the type of mutations and patterns of p53 staining also for LOF variants vs. cases without *TP53* variants (Spearman correlation coefficient = 0.782; *p* < 0.01, Appendix A, Figure 2C–E).

### 3.5. SMAD4 Expression Loss and EAC Survival

We recently reported that SMAD4 loss of immunoreactivity was not an infrequent event in EAC, even in absence of gene mutations [12]. Therefore, we extended the analysis to a larger sample of EAC tissues from the EACSGE consortium and correlated it with genetic, histopathological, and clinical data [7].

First, for the group of cases where we had the *TP53* status and p53 immunostaining, we evaluated whether any correlation with the SMAD4 immunostaining pattern was significant. Indeed, we found that the presence of LOF variants in *TP53* correlated with SMAD4 loss of staining in the corresponding tumor tissue (as defined in the Materials and Method section, *p* = 0.008, Appendix A).

Therefore, we investigated whether any loss of SMAD4 immunostaining was relevant or whether a more informative SMAD4 loss cut-off value could be identified. Through ROC curve analysis (Appendix A), loss of immunostaining in at least 35% of neoplastic cells resulted to be the best discriminator with the greatest sensitivity and specificity. Applying this cut-off value in our case series, 85 cases were defined as SMAD4 loss cancers out of the 245 EAC cases from the entire EACSGE consortium (35%, Figure 3A). When considering the EACSGE histopathological classification, the SMAD4 pattern of immunostaining was significantly correlated with CCS and disease-free survival. SMAD4 loss was correlated with poor CSS (*p* = 0.007; Figure 3B) and disease-free survival (*p* = 0.002, Figure 3C) in EACSGE higher risk cases but not in lower risk cases (*p* = ns).

Post-hoc analysis revealed that the study had 0.772 power to detect a 0.25 effect size [16].

### 3.6. Univariate and Multivariate Cox Regression Analysis

Univariate and multivariate Cox regression analyses were performed to estimate the effects of clinical, genetic (*TP53*), SMAD4 loss (cut-off > 35), and pathological parameters on CSS. As reported in Appendix A, the univariate Cox regression analysis showed a statistical association for age, stage, lymph node status, and EACSGE risk (*p* = 0.028, *p* = 0.001, *p* < 0.001, and *p* = 0.003, respectively), whereas in the multivariate analysis, only age, lymph node ratio, and EACSGE risk retained significance (*p* = 0.005, *p* < 0.001, and *p* = 0.023, Appendix A).

### 3.7. Gene Fusion Analysis from RNA Sequencing

Twenty-two samples, with RNA quality compatible with massive parallel sequencing were sequenced at high coverage for 1385 oncology-relevant genes. Data analysis was performed with an in-house pipeline which is based on the combination of data obtained using four independent tools for fusion detection, e.g., FusionCatcher, STAR-Fusion, RNA-Seq Alignment v.1.1.0, and TopHat Alignment v.1.0.0. From the 64 candidate fusions identified by combining the results of the four tools, the pipeline retained eight after filtering analysis. The selected gene fusions were selected by two to four tools (Appendix A). The eight gene fusions were identified in six different EAC cases, but we could confirm with an independent method (Sanger sequencing) six gene fusions in four cases (4/22, 18.2%, Appendix A). Interestingly, in the two cases carrying two different gene fusions, one of the rearrangements was the same, the *CYP2C19-CYP2C18* fusion on chromosome 10 (Figure 4A). The other gene fusions involved the *GIPC1-DNAJB1* rearrangement in one case (Figure 4B) and *PI4KA-MAPK1* in the other case. The GAIP-interacting protein C-terminus (GIPC1) is a regulator of autophagy and cellular trafficking and its overexpression is associated with poor survival in several cancers [17,18]. *DNAJB1* encodes for a molecular chaperone involved in protein folding and autophagic mechanisms. The *DNAJB1-PRKACA* fusion transcript has been identified in many cases of fibrolamellar hepatocellular carcinoma [19].

The gene encoding for the phosphatidylinositol 4-kinase alpha (*PI4KA*) was detected in two different gene fusion EAC samples (Figure 4C). PI4KA plays a critical role in regulating tumorigenesis by activating tumor-promoting signals such as the RAS pathway [20].

The gene fusion *IQCE-DGKB* (Figure 4D) involved an IQ motif containing E gene, which is important in limb morphogenesis and also acts as regulator of Hedgehog signaling [21] and the gene for diacylglycerol kinase beta (*DGKB*). The diacylglycerol kinases are key regulators of the intracellular concentration of the second messenger diacylglycerol (DAG) and play a key role in cellular processes. This gene has been found in other fusions with different genes in prostate cancer [22].

All of the detected gene fusions involved oncology-related genes, but they have not been reported in esophageal adenocarcinoma.

## 4. Discussion

Esophageal adenocarcinoma represents a substantial health concern in Western countries due to its increasing incidence and poor prognosis. Rapid advances in high-throughput NGS have highlighted high EAC inter- and intratumor heterogeneity, with many structural genomic rearrangements and mutations arising even clonally [8,11,23,24], and epigenetic dysregulation of specific genes giving rise to tumor entities that may behave very differently in terms of progression and resistance [8,25,26]. Therefore, there is an increasing interest in defining molecular biomarkers for patient stratification and prognosis [27].

In our study, we focused our attention on a panel of cancer-related genes previously found recurrently mutated in a small cohort of EAC cases. In the 164 novel cases assessed, the EAC samples presented mutations in different genes, including *HNF1alpha*. HNF1alpha is a transcription factor, regulates epithelial to mesenchymal transition, and is considered a tumor suppressor [28]. In concordance with its role as a tumor suppressor, the majority of the variants were LOF, with a remarkable decrease in protein expression in the tumor specimens compared to cases with no mutations in this gene. *HNF1alpha* mutations were associated with other mutations in most of our cases, as we observed this for several genes, concurrently mutated in the same samples. However, we did not address the overall mutational signatures, in terms of specific nucleotide changes as detected with programs such as Signature Mutational Analysis (Sigma) [29] or Mix [30], but only whether specific genes showed concurrent or mutually exclusive mutations.

Regarding the identified gene mutations, *TP53* was the most frequently mutated gene, as reported in previous studies [8]. Indeed, *TP53* is the most mutated gene in the group of chromosomally unstable carcinomas of the esophagus and of the esophageal junction [8] and tumors that are histologically predominantly intestinal. Thus, it is not surprising that in our classification, they are mainly GD-PD with *TP53* mutations, but it is remarkable that in GD-PD, this finding has prognostic significance. Moreover, we evaluated the effect of specific types of mutations (missense changes vs. loss of function) in correlation with histological and clinical data and could observe that in the cases classified as “higher risk” according to the EACSGE classification, the presence of damaging missense variants in *TP53* negatively affected cancer-specific survival,

It is however worth noting that *TP53* mutations are early events in the progression from esophageal dysplasia to cancer [31]; therefore, the early identification of specific types of variants, i.e., damaging amino acid substitutions, might be of critical importance, especially from the perspective of selecting the most efficient approach for targeted therapies. It is important to note that missense changes in p53 can alter protein folding, structure, and therefore DNA-binding and transcriptional activity. Targeting mutant p53 with missense changes is indeed an active research field in order to induce p53 activity similar to the wild-type protein [32].

Mutations that led to a loss of p53 proteins (such as nonsense and frameshift variants, LOF) were frequently associated with the loss of SMAD4 expression. The *SMAD4* tumor-suppressor gene is pivotal for the downstream signaling of bone morphogenetic proteins (BMPs) [33]. Notably, *SMAD4* has been reported as frequently lost in gastrointestinal cancer [11,34]. SMAD4 loss is associated with non-canonical BMP signaling leading to a more metastatic phenotype, poor prognosis, and poor response to treatment [35]. A previous study showed that *SMAD4* mutations or homozygous deletions were associated with significantly poorer prognosis in EAC [11]. A recent study in preclinical models of EAC development showed that *SMAD4* inactivation was sufficient per se to initiate tumorigenesis in a high-grade dysplastic esophagus in vivo [36]. Moreover, inhibition of the overactive non-canonical BMP signaling in SMAD4-negative tumors decreased malignancy and improved survival [37]. Notably, SMAD4 expression status resulted in being a prognostic factor in our cohort of cases when connected to the EACSGE classification, allowing us to discriminate patients with a worse prognosis into the higher risk group.

Nevertheless, when using multivariate Cox regression analyses for CSS and different variables, these associations with *TP53* or *SMAD4* status were not so well-defined, whereas age, lymph node status, and EACSGE risk still retained a significant correlation. Therefore, further studies in independent and large samples are warranted in order to evaluate the clinical–pathological correlation with specific types of *TP53* mutations and SMAD4 expression.

A limit of our study is that on the tumor DNA extracted from paraffin-embedded tissue biopsies, we performed a targeted analysis of a discrete number of oncology-related genes and did not perform a whole exome or whole genome analysis. Thus, we were not able to also evaluate the presence of copy number alterations (CNAs), because our target gene panel was designed to test for single nucleotide or small insertion/deletion variants. Nevertheless, we observed a number of gene fusion transcripts (18.2%) using a high throughput RNA sequencing approach involving oncology-related genes. Gene fusion products can become ideal therapeutic targets, as observed in other cancers [34,38]. The identified gene fusions have not been reported previously in EAC and in three cases out of the four carrying the identified gene fusions, *TP53* was mutated. Although the number of cases for which RNA was available was too small to drive statistically significant conclusions, we suggest that this analysis would add an additional step toward understanding the molecular complexity of EAC, in accordance with other studies investigating the structural rearrangements in EAC [39].

In this view, we suggest that the investigation of molecular markers together with a histopathological analysis can provide relevant clinical information for patient stratification, treatment, and prognosis. The development of therapies targeting specific pathways, based on the status of molecular biomarkers, will improve EAC clinical management.

## 5. Conclusions

In our study on EAC, we were able to correlate EAC histological classification, clinical outcomes, and molecular phenotypes. First, we identified loss-of-function mutations in *HNF1alpha*, a tumor suppressor gene with a likely role in tumor progression. Next, we showed that *TP53* missense mutations are associated with higher risk cases, and in this sub-group, they could contribute to a poorer outcome (CSS). *TP53* truncating mutations were associated with SMAD4 loss, and SMAD4 loss itself was a frequent event in EAC, correlated with lower CSS and disease-free survival in higher risk cases. Therefore, we showed that combining molecular and histological analyses could be a successful strategy to better stratify patients. This could contribute to identifying those patients with a worse prognosis and to selecting a tailored therapy based on molecular markers.

## Figures and Tables

**Figure 1 cancers-15-01408-f001:**
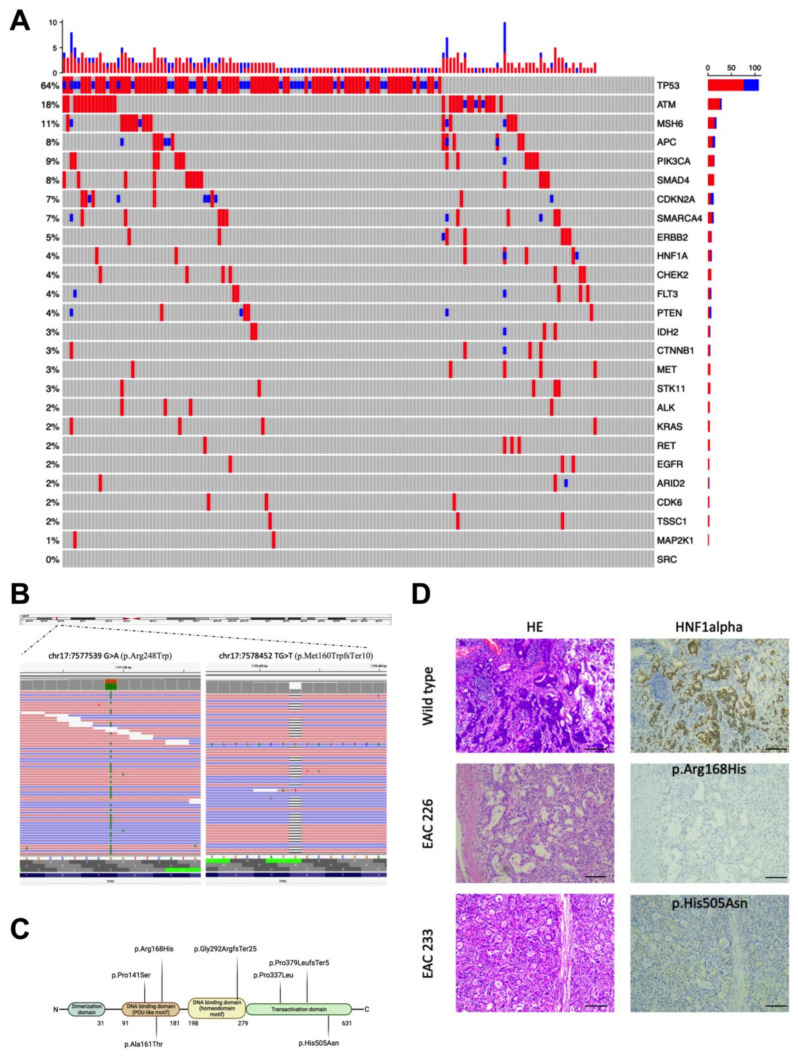
Gene variants identified in 164 EAC cases. (**A**) Variants identified in the EAC cohort; in red “missense”; in blue “LOF (frameshift and stop codon)”. (**B**) Representation of two *TP53* mutations (visualized using Integrative Genomic Viewer, IGV): a missense (p.Arg248Trp) and a frameshift (p.Met160TrpfsTer10). The display shows individual forward (F) sequence reads in red and reverse (R) reads in blue. Selected positions are covered by a high number of aligned sequenced reads and both mutations can be seen in approximately half of the F and R reads. (**C**) Mutations in the *HNF1alpha* gene were identified in the EAC samples and mapped to the different protein domains. (**D**) Examples of the immunohistochemical patterns observed in a control sample (i.e., no *HNF1alpha* mutations) vs. cases carrying different alterations in the *HNF1alpha* gene. In particular, we showed the expression pattern in two cases with the *HNF1alpha* p.Arg168His variant or the p.His505Asn; the decreased staining suggested that the misfolded proteins might be degraded. Scale bar 500 μm.

**Figure 2 cancers-15-01408-f002:**
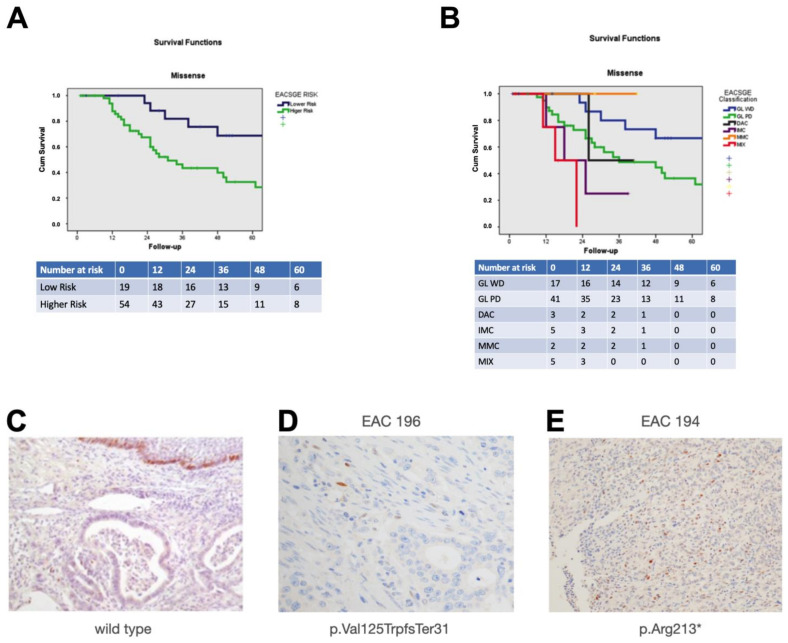
Cancer-specific survival of EAC cases with *TP53* missense variants and p53 expression profiles. (**A**) Data are shown according to the higher and lower risk groups. (**B**) Data are shown for the EACSGE morphological subgroups. (**C**–**E**) Immunohistochemical patterns for p53 immunostaining in the EAC cases with different types of variants of *TP53*: (**C**) control case with no variant in *TP53* vs. loss of function (LOF) (**D**,**E**). Scale bar 100 μm.

**Figure 3 cancers-15-01408-f003:**
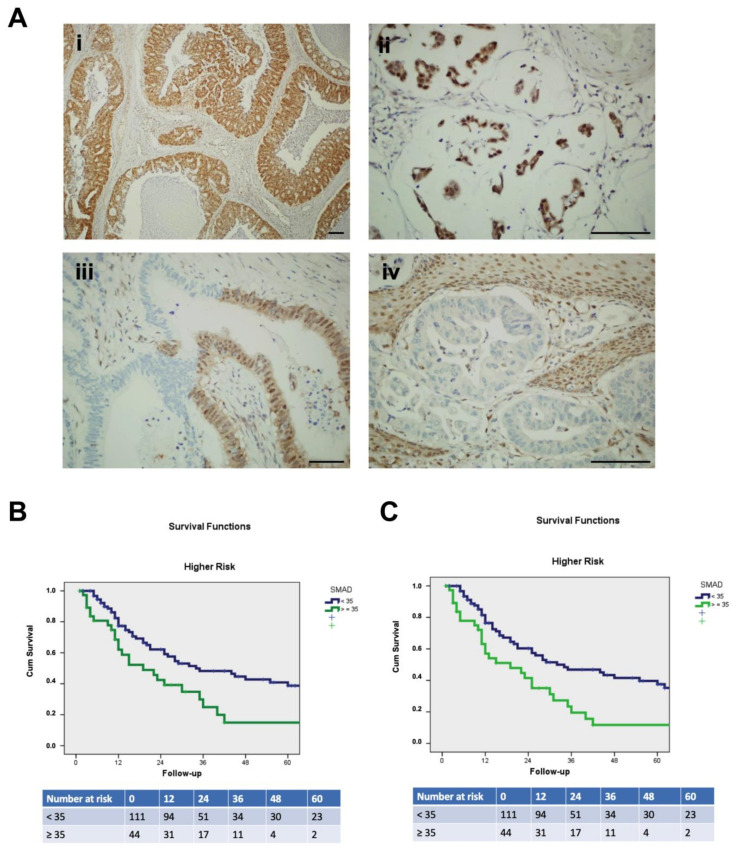
SMAD4 expression patterns and correlation with survival. (**A**) Immunohistochemical patterns observed for SMAD4 expression. Intense preserved SMAD4 immunostain in a sample of differentiated glandular adenocarcinoma (**i**) and in a sample of mucinous muconodular adenocarcinoma (**ii**). A sample of glandular adenocarcinoma with preserved (right side) and complete loss of SMAD4 expression in the same neoplastic gland (**iii**). Complete loss of SMAD4 in glandular adenocarcinoma with retained expression in the squamous epithelium overlying cancer (**iv**). Scale bar, 100 μm. (**B**) Cancer-specific survival and (**C**) disease-free survival of EAC cases with loss of SMAD4 staining (in >35% of the neoplastic area) in EACSGE higher risk cases.

**Figure 4 cancers-15-01408-f004:**
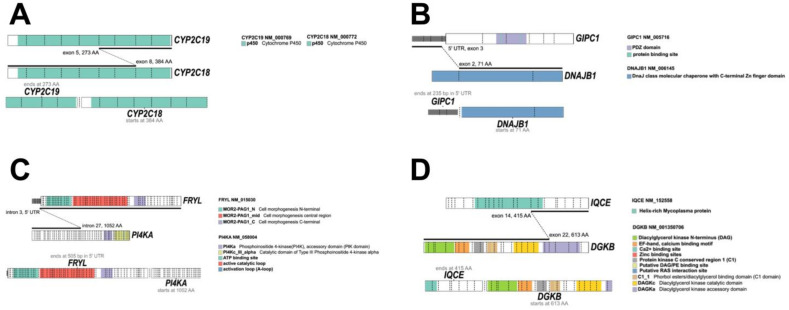
Reconstruction of the different gene fusions identified and validated in the EAC samples. The breakpoints in the gene transcripts and corresponding protein regions and domains are indicated for the genes involved in each gene fusion (**A**–**D**).

## Data Availability

All data are available from the authors upon request.

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
