# Peer review of "Correlations between Molecular Alterations, Histopathological Characteristics, and Poor Prognosis in Esophageal Adenocarcinoma"

_cancers, 2023, doi:10.3390/cancers15051408_

Round 1
Reviewer 1 Report
This interesting study involves the correlation of genetic sequencing information from selected known mutated genes in gastroesophageal adenocarcinoma with survival. The study involved a total of 164 samples from patients who were treatment naïve. Patients had their pathology classified according to new EACSGE standards. Paraffin fixed tissues had DNA extracted for next generation sequencing of known genetic drivers of adenocarcinoma. A selected subset of 5 patients from each histological adenocarcinoma subclassification were studied with RNA sequencing. Immunohistochemistry was performed to examine selected mutations such as HNF1a and SMAD4. Results were correlated to patient medical histories and disease specific survival. This study has several strong suits being one of the first to correlate the genomics and survival and deserves to be published. There are a few minor concerns that should be addressed.
1) Figure 1 illustrates that 26 genes were investigated in this study, given the 164 samples involved, the authors should state the power of this study to demonstrate any
significant association with survival given the large number of genes investigated.
Overall, I believe this study is underpowered to find associations and readers should be
informed of the likelihood that the lack of correlations is likely due to the lack of power.
2) Figure 1, it is difficult to actually see the number of cases of missense and frameshift
mutations in TP53 given the small size of the figure, given the large numbers of large
repeats present in TP53, the authors should provide more details on how alignment was
performed to ensure these mutations were not the result of errors of alignment of the
fragments after sequencing.
3) Figure 1, although two expert pathologists were used to score IHC of HIFN1 as shown in
the representative slides, it would enhance the paper to use digital image analysis (ie
nanostring) to actually quantify staining and also determine where the targets are
located within the tumor.
4) Correlating patterns of mutations were performed using Pearson’s coefficient. It might
be worthwhile using the COSMIC Mutational Signatures database to determine if there
are patterns of mutations that are present that have already been identified or are
these actually unique to esophageal adenocarcinoma.
5) The correlation of histological subtypes in conjunction with mutations to survival is quite
interesting. It is stated this was done with clinical information. However, given that this
is a survival study, there is no data presented regarding tumor staging (either clinical or
pathological) which would help to determine how subtype information and mutational
data added to the survival based on staging which would be helpful clinically.
6) Given that only 22 samples were assessed with RNA Seq, the higher rate of fusions
found in this cohort of patients is interesting though possibly confounded by the small
numbers. I would encourage the authors to be a cautious in their interpretations
regarding the prevalence of fusions given their small sample size. In addition, the
investigators only examined specific genes rather than GWAS and transcriptomics.
Author Response
Reviewer 1
Comments and Suggestions for Authors
This interesting study involves the correlation of genetic sequencing information from selected known mutated genes in gastroesophageal adenocarcinoma with survival. The study involved a total of 164 samples from patients who were treatment naïve. Patients had their pathology classified according to new EACSGE standards. Paraffin fixed tissues had DNA extracted for next generation sequencing of known genetic drivers of adenocarcinoma. A selected subset of 5 patients from each histological adenocarcinoma subclassification were studied with RNA sequencing. Immunohistochemistry was performed to examine selected mutations such as HNF1a and SMAD4. Results were correlated to patient medical histories and disease specific survival. This study has several strong suits being one of the first to correlate the genomics and survival and deserves to be published. There are a few minor concerns that should be addressed.
1) Figure 1 illustrates that 26 genes were investigated in this study, given the 164 samples involved, the authors should state the power of this study to demonstrate any
significant association with survival given the large number of genes investigated.
Overall, I believe this study is underpowered to find associations and readers should be
informed of the likelihood that the lack of correlations is likely due to the lack of power.
We are grateful to the reviewer for this important comment. In order to assess the power of the study we performed a post-hoc analysis. The text was modified in the statistical method and results sections, according to the reviewer’s suggestions.
For power calculations we used G*Power version 3.1.9.6 [Faul F, Erdfelder E, Lang A-G, et al.. G*Power 3: a flexible statistical power analysis program for the social, behavioral, and biomedical sciences. Behav Res Methods 2007;39:175–91. 10.3758/BF03193146] now new reference#14.
Post-hoc analysis revealed that the study had 0.772 power to detect 0.25 effect size [Cohen J. A power primer. Psychol Bull 1992;112:155–9. 10.1037/0033-2909.112.1.155]now new reference #16, .
The text has been modified as following:
Methods: page 4, line 186: “For power calculations we used G*Power version 3.1.9.6 [14].”
Results: page 7, line 285: “Post-hoc analysis revealed that the study had 0.772 power to detect 0.25 effect size [16].”
2) Figure 1, it is difficult to actually see the number of cases of missense and frameshift
mutations in TP53 given the small size of the figure, given the large numbers of large
repeats present in TP53, the authors should provide more details on how alignment was
performed to ensure these mutations were not the result of errors of alignment of the
fragments after sequencing.
We modified Figure 1, inserting a new panel B showing the coverage of detected mutation in IGV and added more details on coverage, alignment and data interpretation. The text has been modified as follows:
Methods: page 3, lines 146-152. “In particular, Fastq files containing raw reads were checked using FastQC (https://www.bioinformatics.babraham.ac.uk/projects/fastqc/) and aligned using BWA (bio-bwa.sourceforge.net) to the human reference (hg19). PCR-duplicated reads were marked and removed using Picard. Putative somatic variants, including SNPs and small insertions/deletions (indels), were identified using GATK software (soft-ware.broadinstitute.org/gatk/). The raw mutation calls were filtered to exclude false calls based on base quality, allele frequency of mismatched bases and possible occurrences of strand bias. The identified mutations were further annotated and prioritized with Ensembl VEP (www.ensembl.org/Tools/VEP).”
Results: page 4, lines 190-192. “All FFPE samples achieved good sequence representation with average coverage among samples of 700X. Examples of identified mutations are reported in Figure 1B.”
3) Figure 1, although two expert pathologists were used to score IHC of HIFN1 as shown in
the representative slides, it would enhance the paper to use digital image analysis (ie
nanostring) to actually quantify staining and also determine where the targets are
located within the tumor.
We thank the reviewer for the suggestion, but we do not have yet Nanostring or single cell-RNAseq technology currently available. It will be of great interest in future works to apply these technologies, however the IHC staining was performed according to previously published works by our group (references 12 and 13).
4) Correlating patterns of mutations were performed using Pearson’s coefficient. It might
be worthwhile using the COSMIC Mutational Signatures database to determine if there
are patterns of mutations that are present that have already been identified or are
these actually unique to esophageal adenocarcinoma.
We apologize for the mistake, but the correlation analysis was performed with the Spearman’s correlation coefficient. We amended the mistakes in the main text and supplementary tables. We also evaluated the COSMIC database and inserted the description of findings in the text as follows:
Results: page 6, lines 228-231. “specific genes were concurrently mutated in these tumors. We also assessed in the COSMIC project COSU535, containing data on 409 EAC cases, the presence of co-occurring variants between genes and found mutations of TP53 and APC (21/409, 5%), TP53 and CDKN2A in 25/409 (6%), and TP53 and SMAD4 in 30/409 (7%) in the same samples, reinforcing the concept of the genetic heterogeneity of EAC.”
5) The correlation of histological subtypes in conjunction with mutations to survival is quite
interesting. It is stated this was done with clinical information. However, given that this
is a survival study, there is no data presented regarding tumor staging (either clinical or
pathological) which would help to determine how subtype information and mutational
data added to the survival based on staging which would be helpful clinically.
We thank the reviewer for this helpful suggestion. Therefore, we performed a Cox univariate and multivariate analysis evaluating among the variables also the pathological stage, according to the AJCC 8th edition and the ratio of metastatic lymph nodes to the total number of nodes yielded, rather than the parameter N, already considered in the pathological stage. In fact, in a previous study of our group we had found that the metastatic lymph nodes/lymph nodes yielded ratio and not the total number of lymph nodes yielded, did correlate with cancer-specific survival (for a reference please see: Ann Thorac Surg. 2016 May;101(5):1915-20).
We modified the text according to the reviewer’s suggestion and we added the Supplementary Table 8.
Methods: page 4, lines 183-186. “Univariate and multivariate (forward stepwise conditional method) Cox regression analyses were performed to estimate the effects of clinical, genetic and pathological parameters on CSS. In the stepwise procedure, significance levels of 0.05 for entering and 0.10 for removing the respective explanatory variables were used to determine the independent risk factors.”
Results: page 8, lines 289-294. “3.5 Univariate and multivariate Cox regression analysis
Univariate and multivariate Cox regression analyses were performed to estimate the effects of clinical, genetic (TP53), SMAD4 loss (cut-off >35) and pathological parameters on CSS. As reported in Supplementary Table 8A, univariate Cox regression analysis showed a statistical association for age, stage, lymph node status and EACGSE risk (P=0.028, P=0.001, P=10-3, P=0.003, respectively), whereas in multivariate analysis only age, lymph node ratio and EACGSE risk retained significance (P=0.005, P<0.001, P=0.023, Supplementary Table 8B).”
Discussion: page 10, lines 365-368. “Nevertheless, when using multivariate Cox regression analyses for CSS and different variables, these associations with TP53 or SMAD4 status were not so well-defined, whereas age, lymph node status and EACGSE risk still re-tained a significant correlation. Therefore, further studies in independent and large samples should be warranted in order to evaluate the clinical-pathological correlation with specific types of TP53 mutations and with SMAD4 expression.”
6) Given that only 22 samples were assessed with RNA Seq, the higher rate of fusions
found in this cohort of patients is interesting though possibly confounded by the small
numbers. I would encourage the authors to be a cautious in their interpretations
regarding the prevalence of fusions given their small sample size. In addition, the
investigators only examined specific genes rather than GWAS and transcriptomics.
We thank the reviewer for this comment. Regarding genome wide association studies (GWAS) any analysis would require many more samples for deriving any statistical association and usually is not performed on FFPE-extracted DNA, but on constitutive genomic DNA for SNP analysis. For whole transcriptomics, the quality and quantity of the RNA extracted from the same paraffin-embedded samples for which we performed target gene analysis, was not suitable to proceed to whole transcriptome analysis. Ideally, fresh tissues collected in a prospective study would be a better source of material for these types of studies. Nevertheless, for a small set of samples we were able to perform RNAseq on >1000 oncology-related genes with high coverage and detected previously unreported gene fusions. Indeed we did want to be extremely cautious in defining the role of these gene fusions in EAC, and modified the text accordingly.
Discussion: page 10, lines 369-376. “A limit of our study is that on the tumor DNA extracted from paraffin-embedded tissue biopsies we performed a targeted analysis of a discrete number of oncology-related genes and did not perform a whole exome or whole genome analysis. Thus, we were not able to evaluate also the presence of-Copy Number Alterations (CNAs), because our target gene panel was designed to test for single nucleotide or small insertion/deletion variants. Nevertheless, we observed a number of gene fusion transcripts (18.2%) using a high throughput RNA sequencing approach, involving oncology-related genes.”
Reviewer 2 Report
The authors show interesting data regarding molecular alterations in EAC. I think some revision would contribute to the value of this paper for the scientific community.
The authors connect their findings to some the EACSGE classification. The UICC stage and or the pTNM of the analysed tumor samples should also be reported.These genomic data are very valuable and contribute to knowledge in the field. I think it is crucial for our understand of this disease how molecular alterations and e.g. the lymphonodal status relate to each other. Especially, as the authors report survival data, the lymphonodal status cannot be ignored. If the authors believe that HNF alpha damaging variants contribute to tumor progression, it would be interesting to see the pT and pN status of these specific samples. The homogenous treatment naïve cohort the authors present is of value here.
I would suggest performing a cox regression analysis for survival data and not only a long-rank tests. These should also include multivariable analyses considering the UICC stage or at least the pN status.
Minor:
line 246: maybe change the spelling of the p-value to 0.001 to be more consistent.
line 252: cancer-specific survival
line 255: categories
Figure 2: Addition of a risk table would provide a better overview about the number of cases in
Figure 3: Add risk tables, also add title to each KM figure.
Author Response
Reviewer 2
The authors show interesting data regarding molecular alterations in EAC. I think some revision would contribute to the value of this paper for the scientific community.
The authors connect their findings to some the EACSGE classification. The UICC stage and or the pTNM of the analysed tumor samples should also be reported.These genomic data are very valuable and contribute to knowledge in the field. I think it is crucial for our understand of this disease how molecular alterations and e.g. the lymphonodal status relate to each other. Especially, as the authors report survival data, the lymphonodal status cannot be ignored. If the authors believe that HNF alpha damaging variants contribute to tumor progression, it would be interesting to see the pT and pN status of these specific samples. The homogenous treatment naïve cohort the authors present is of value here.
We thank the reviewer for the comment. The pT and pN status of samples mutated in HNF1 alpha were added in the last two columns of Supplementary Table 2.
I would suggest performing a cox regression analysis for survival data and not only a long-rank tests. These should also include multivariable analyses considering the UICC stage or at least the pN status.
We thank the reviewer for this helpful suggestion. As indicated in the answer #5 of reviewer 1, we performed a Cox univariate and multivariate analysis evaluating among the variables also the pathological stage, according to the AJCC 8th edition and the ratio of metastatic lymph nodes to the total number of nodes yielded, rather than the parameter N, already considered in the pathological stage. In fact, in a previous study of our group we had found that the metastatic lymph nodes/lymph nodes yielded ratio and not the total number of lymph nodes yielded, did correlate with cancer-specific survival (Ann Thorac Surg. 2016 May;101(5):1915-20).
We modified the text according to the reviewer’s suggestion and we added the Supplementary Table 8.
Methods: page 4, lines 183-186. “Univariate and multivariate (forward stepwise conditional method) Cox regression analyses were performed to estimate the effects of clinical, genetic and pathological parameters on CSS. In the stepwise procedure, significance levels of 0.05 for entering and 0.10 for removing the respective explanatory variables were used to determine the independent risk factors.”
Results: page 8, lines 289-294. “3.5 Univariate and multivariate Cox regression analysis
Univariate and multivariate Cox regression analyses were performed to estimate the effects of clinical, genetic (TP53), SMAD4 loss (cut-off >35) and pathological parameters on CSS. As reported in Supplementary Table 8A, univariate Cox regression analysis showed a statistical association for age, stage, lymph node status and EACGSE risk (P=0.028, P=0.001, P=10-3, P=0.003, respectively), whereas in multivariate analysis only age, lymph node ratio and EACGSE risk retained significance (P=0.005, P<0.001, P=0.023, Supplementary Table 8B).”
Discussion: page 10, lines 365-368. “Nevertheless, when using multivariate Cox regression analyses for CSS and different variables, these associations with TP53 or SMAD4 status were not so well-defined, whereas age, lymph node status and EACGSE risk still re-tained a significant correlation. Therefore, further studies in independent and large samples should be warranted in order to evaluate the clinical-pathological correlation with specific types of TP53 mutations and with SMAD4 expression.”
Minor:
line 246: maybe change the spelling of the p-value to 0.001 to be more consistent.
We modified the text accordingly
line 252: cancer-specific survival
Corrected
line 255: categories
Corrected
Figure 2: Addition of a risk table would provide a better overview about the number of cases in
Figure 3: Add risk tables, also add title to each KM figure.
We added titles and the risk tables to each corresponding KM in Figure 2 and in Figure 3.
Reviewer 3
The authors show interesting data regarding molecular alterations in EAC. I think some revision would contribute to the value of this paper for the scientific community.
The authors connect their findings to some the EACSGE classification. The UICC stage and or the pTNM of the analysed tumor samples should also be reported.These genomic data are very valuable and contribute to knowledge in the field. I think it is crucial for our understand of this disease how molecular alterations and e.g. the lymphonodal status relate to each other. Especially, as the authors report survival data, the lymphonodal status cannot be ignored. If the authors believe that HNF alpha damaging variants contribute to tumor progression, it would be interesting to see the pT and pN status of these specific samples. The homogenous treatment naïve cohort the authors present is of value here.
We thank the reviewer for the comment. The pT and pN status of samples mutated in HNF1 alpha were added in the last two columns of Supplementary Table 2.
I would suggest performing a cox regression analysis for survival data and not only a long-rank tests. These should also include multivariable analyses considering the UICC stage or at least the pN status.
We thank the reviewer for this helpful suggestion. As indicated in the answer #5 of reviewer 1 and in answer #2 of reviewer 2, we performed a Cox univariate and multivariate analysis evaluating among the variables also the pathological stage, according to the AJCC 8th edition and the ratio of metastatic lymph nodes to the total number of nodes yielded, rather than the parameter N, already considered in the pathological stage. In fact, in a previous study of our group we had found that the metastatic lymph nodes/lymph nodes yielded ratio and not the total number of lymph nodes yielded, did correlate with cancer-specific survival.
We modified the text according to the reviewer’s suggestion and we added the Supplementary Table 8.
Methods: page 4, lines 183-186. “Univariate and multivariate (forward stepwise conditional method) Cox regression analyses were performed to estimate the effects of clinical, genetic and pathological parameters on CSS. In the stepwise procedure, significance levels of 0.05 for entering and 0.10 for removing the respective explanatory variables were used to determine the independent risk factors.”
Results: page 8, lines 289-294. “3.5 Univariate and multivariate Cox regression analysis
Univariate and multivariate Cox regression analyses were performed to estimate the effects of clinical, genetic (TP53), SMAD4 loss (cut-off >35) and pathological parameters on CSS. As reported in Supplementary Table 8A, univariate Cox regression analysis showed a statistical association for age, stage, lymph node status and EACGSE risk (P=0.028, P=0.001, P=10-3, P=0.003, respectively), whereas in multivariate analysis only age, lymph node ratio and EACGSE risk retained significance (P=0.005, P<0.001, P=0.023, Supplementary Table 8B).”
Discussion: page 10, lines 365-368. “Nevertheless, when using multivariate Cox regression analyses for CSS and different variables, these associations with TP53 or SMAD4 status were not so well-defined, whereas age, lymph node status and EACGSE risk still re-tained a significant correlation. Therefore, further studies in independent and large samples should be warranted in order to evaluate the clinical-pathological correlation with specific types of TP53 mutations and with SMAD4 expression.”
Minor:
line 246: maybe change the spelling of the p-value to 0.001 to be more consistent.
We modified the text accordingly
line 252: cancer-specific survival
Corrected
line 255: categories
Corrected
Figure 2: Addition of a risk table would provide a better overview about the number of cases in Figure 3: Add risk tables, also add title to each KM figure.
We added titles and the risk tables to each corresponding KM in Figure 2 and in Figure 3.
Author Response
In this study, Orsini, Bozzarelli, Mastracci et al. present a correlative study between the molecular make-up, histopathological features and prognosis data in 164 treatment naive esophageal adenocarcinoma (EAC) cases. The aim of the study was to integrate these features in order to find a ‘biomarker’ that may aid in clinical management of EAC patients. To that effect. the data from a targeted sequencing panel and histopathological features were obtained and correlated. Additionally RNA-seq was performed on a subset of the cases to detect gene fusions as a proxy to genomic instability. I have some critiques about the analyses of the data and the conclusions that were drawn and are as follows:
Major Critiques:
- Overall there is a lack of robustness as far as the correlative analyses is concerned. For the survival analyses concerning the presence of TP53 variants, a simple Kaplan-Meier curve is insufficient to draw any meaningful conclusion. There are several confounding features that may have to be corrected to see if the difference in the survival is due to the presence of mutations alone. An obvious confounder that comes to mind is age.
This point is also relevant to the top level EACGSE (low and high risk) classification in TP53 mutant cases (fig. 2A). The authors themselves point out that the signal was predominantly driven by GD-PD class. This further warrants using a more robust strategy. I would suggest using a model such as the Cox proportional hazards model for these analyses.
The effect of age was assessed in the analysis, the text was modified in the section results and data shown in the novel Supplementary Table 5C.
Results: page 6, lines 257-258. “TP53 mutations and age also showed a significant association, as reported in Supplementary Table 5C (P=0.029, Kruskal-Wallis test).”
With regards to Cox analysis, as indicated in the answer #5 of reviewer 1 and in answer #2 of reviewers 2 and 3, we performed a Cox univariate and multivariate analysis evaluating among the variables also the pathological stage, according to the AJCC 8th edition and the ratio of metastatic lymph nodes to the total number of nodes yielded, rather than the parameter N, already considered in the pathological stage. In fact, in a previous study of our group we had found that the metastatic lymph nodes/lymph nodes yielded ratio and not the total number of lymph nodes yielded, did correlate with cancer-specific survival.
We modified the text according to the reviewer’s suggestion and we added the Supplementary Table 8.
Methods: page 4, lines 183-186. “Univariate and multivariate (forward stepwise conditional method) Cox regression analyses were performed to estimate the effects of clinical, genetic and pathological parameters on CSS. In the stepwise procedure, significance levels of 0.05 for entering and 0.10 for removing the respective explanatory variables were used to determine the independent risk factors.”
Results: page 8, lines 289-294. “3.5 Univariate and multivariate Cox regression analysis
Univariate and multivariate Cox regression analyses were performed to estimate the effects of clinical, genetic (TP53), SMAD4 loss (cut-off >35) and pathological parameters on CSS. As reported in Supplementary Table 8A, univariate Cox regression analysis showed a statistical association for age, stage, lymph node status and EACGSE risk (P=0.028, P=0.001, P=10-3, P=0.003, respectively), whereas in multivariate analysis only age, lymph node ratio and EACGSE risk retained significance (P=0.005, P<0.001, P=0.023, Supplementary Table 8B).”
Discussion: page 10, lines 365-368. “Nevertheless, when using multivariate Cox regression analyses for CSS and different variables, these associations with TP53 or SMAD4 status were not so well-defined, whereas age, lymph node status and EACGSE risk still re-tained a significant correlation. Therefore, further studies in independent and large samples should be warranted in order to evaluate the clinical-pathological correlation with specific types of TP53 mutations and with SMAD4 expression.”
- The use of Pearsons’s correlation to evaluate the presence of ‘mutation signatures
occurring in specific genes’ is not a sound approach. There are methods that can be
deployed for sparse targeted sequencing that use probabilistic models to produce
signatures (PMID: 34724984). If this is not feasible, the authors should either not include
correlative signature analyses or address the shortcoming in discussion.
We are truly sorry for the wording mistake, but indeed we adopted the Spearman's correlation coefficient and the text was consequently corrected.
Results: page 6, lines 222-223. “we performed a correlation analysis between the damaging variants identified in the different oncology-related genes, using the Spearman’s rho coefficient.”
We also amended the text to clarify our scope in this analysis, since we did not perform a mutational signature analysis (i.e. specific change in the nucleotide such as C to T etc) analysis, as indicated by the method suggested by the reviewers, but investigated whether genes were concurrently mutated Therefore, we modified the text accordingly in the Discussion, to clarify this issue, and cited the indicated reference of the program Mix.
Discussion: page 9, lines 337-339. “However, we did not address the overall mutational signatures, in term of specific nucleotide changes, as detected with programs such as Signature Mutational Analysis (Sigma) [29], or Mix [30], but only whether specific genes showed concurrent or mutually exclusive mutations.”
The lines 247:249 warrant this change.
We modified the text accordingly
- Point (1) is applicable to the SMAD4 loss of immunoreactivity analyses.
We thank the reviewer for the suggestion and would like to indicate that univariate and multivariate COX analyses were performed considering also SMAD4 immunoreactivity as one of the variables. Please refer to answer to Point 1 for the modification throughout the text of the manuscript.
- Presence of HNF1alpha LOF mutations in 7 samples may not be enough to draw
conclusion of the significance of this finding. The authors need to add this caveat in the
discussion. Mere correlation with other mutations is not enough and additional evidence
will be needed to claim that ‘HNF1alpha damaging variants might contribute to tumor
progression in EAC’.
We thank the reviewer for the insightful comment. The phrase on HNF1alpha has now been removed from the text in the Discussion.
- Although this is a targeted panel, there is still scope to characterize copy number
alterations for these relevant genes. Did the authors investigate copy number alteration
landscape in these patients? If there were no significant findings, please mention that in
the discussion.
We thank the reviewer for the comment. The target custom gene panel from IDT was not designed for calling copy number variants, but mainly single nucleotide and small indel variants, therefore we could not apply CNA analysis to our samples. We added the discussion of this limit of the study in the Discussion, as follows:
Page 10, lines 369-376. “A limit of our study is that on the tumor DNA extracted from paraffin-embedded tissue biopsies we performed a targeted analysis of a discrete number of oncology-related genes and did not perform a whole exome or whole genome analysis. Thus, we were not able to evaluate also the presence of Copy Number Alterations (CNAs), because our target gene panel was designed to test for single nucleotide or small insertion/deletion variants. Nevertheless, we observed a number of gene fusion transcripts (18.2%) using a high throughput RNA sequencing approach, involving oncology-related genes.”
Minor Critiques:
- Please include a scale for all the histo-pathological slides.
Done.
- Citations no. 15 and 16 are not included in the text. Please make sure they are used in
the relevant places.
We carefully rechecked the submitted version of the manuscript and actually found that the references 15 and 16 were not correctly indicated. We have now amended the references in the revised version. There are now the new references 17 and 18, as follows:
References: page 12, lines 456-460. “17. Chittenden, T.W., Pak, J., Rubio, R., Cheng, H., Holton, K., Prendergast, N., Glinskii, V., Cai,Y., Culhane, A., Bentink, S., et al. Therapeutic implications of GIPC1 silencing in cancer. PLoS One. 2010, 5, e15581. doi: 10.1371/journal.pone.0015581.
- Katoh M. Functional proteomics, human genetics and cancer biology of GIPC family members. Exp Mol Med. 2013, 45, e26. doi: 10.1038/emm.2013.49.”
3. Orphan parenthesis on line 116.
Removed, thank you
Round 2
Reviewer 3 Report
No further comments to the authors.